# A Hardware Security Protection Method for Conditional Branches of Embedded Systems

**DOI:** 10.3390/mi15060760

**Published:** 2024-06-05

**Authors:** Qiang Hao, Dongdong Xu, Yusen Qin, Ruyin Li, Zongxuan Zhang, Yunyan You, Xiang Wang

**Affiliations:** School of Electronic and Information Engineering, Beihang University, Beijing 100191, China; haoqiang1994@buaa.edu.cn (Q.H.); xudongdong1994@buaa.edu.cn (D.X.); qinys@buaa.edu.cn (Y.Q.); ruyinli@buaa.edu.cn (R.L.); zzxuan@buaa.edu.cn (Z.Z.); yyy614@buaa.edu.cn (Y.Y.)

**Keywords:** embedded system, branch prediction unit, conditional branch, jump address, jump direction

## Abstract

The branch prediction units (BPUs) generally have security vulnerabilities, which can be used by attackers to tamper with the branches, and the existing protection methods cannot defend against these attacks. Therefore, this article proposes a hardware security protection method for conditional branches of embedded systems. This method calculates the number of branch target buffer (BTB) updates every 80 clock cycles. If the number exceeds the set threshold, the BTB will be locked and prevent any process from tampering with the BTB entries, thereby resisting branch prediction analysis (BPA) attacks. Moreover, to prevent attackers from stealing the critical information of branches, the method designs the hybrid arbiter physical unclonable function (APUF) circuit to encrypt and decrypt the directions, addresses, and indexes of branches. This circuit combines the advantages of double APUF and Feed-Forward APUF, which can enhance the randomness of output response and resist machine learning attacks. If attackers still successfully tamper with the branches and disrupt the control flow integrity (CFI), this method detects tampering with the instruction codes, jump addresses, and jump directions in a timely manner through dynamic and static label comparison. The proposed method is implemented and tested on FPGA. The experimental results show that this method can achieve fine-grained security protection for conditional branches, with about 5.4% resource overhead and less than 5.5% performance overhead.

## 1. Introduction

Most existing embedded systems use branch prediction units (BPUs) to predict the jump addresses and jump directions of conditional branches, which can improve the task processing speed [1]. However, with the discovery of the Spectre and Meltdown vulnerabilities [2,3], these high-performance processor architectures have the security risk of sensitive data leakage. Attackers can inject malicious code, steal confidential information, tamper with critical data, and compromise the control flow integrity (CFI) of a program through these vulnerabilities that exist in the BPUs [4]. Therefore, embedded system designers should consider security issues at the chip architecture level and adopt secure and efficient protection methods to address these security vulnerabilities [5]. Branch History Table (BHT) and Branch target buffer (BTB) are important modules of the BPU. BHT is used to predict whether the conditional branches will be taken, that is, to predict the jump directions of branches. BTB is used to predict the jump addresses of conditional branches. Attackers can analyze the data of BHT and BTB to obtain the prior knowledge required for constructing malicious code and stealing sensitive information [6,7]. To defend against these attacks, various software-based or hardware-based security protection methods have been proposed [8,9]. Compared with software-based methods, hardware-assisted methods use fewer resources and have faster speed [10]. Therefore, our research is based on a hardware-assisted security protection method. Members of our lab have previously proposed various hardware security protection methods for embedded systems [11,12,13,14,15], but when applied, it was found that attackers could still exploit the vulnerabilities of BPUs to perform attacks. Therefore, we began to refer to existing methods [16,17,18,19,20].

Attackers can use branch prediction analysis (BPA) attacks [20] to determine whether a branch has been executed. The characteristic of BPA attacks is to fill the BTB as much as possible through the spy processes. Therefore, a method was proposed [20], which detected the spy processes by monitoring the occupancy rate of BTB during program execution and prevented new data from being written to the BTB for a period of time thereafter. However, when we tried to apply this method to lightweight embedded processors, we found that if the structure of BTB was relatively small (such as the Xuan Tie E906 with only 16 BTB entries), even without BPA attacks, the processor would still have a high occupancy rate of BTB when executing programs. Therefore, existing protection methods sometimes struggle to correctly distinguish between normal processes and spy processes.

To prevent attackers from stealing critical information, researchers have proposed many confidentiality protection methods. Ref. [17] proposed a secure branch predictor that encrypted the content and indexes of the BTB and BHT. Ref. [19] proposed a method that used PUF to encrypt return addresses and jump addresses. Inspired by existing methods, we attempted to use PUF circuits to encrypt and decrypt the critical information of branches. However, some references pointed out that PUF circuits had difficulty resisting machine learning (ML) attacks [21]. Moreover, we attempted to use the higher security cryptographic algorithms [11,12,22], but the performance and resource overhead of these cryptographic circuits were significant, which made it difficult to achieve confidentiality protection for branch information in lightweight embedded processors.

Researchers have proposed various hardware protection methods to monitor the CFI during program execution. Ref. [10] tried to restrict the target address of the jump instruction to the starting address of a basic block (BB), and this method could defend more than 90% of attacks against the jump instructions, but its performance overhead was more than 9%. Ref. [13] determined whether the control flow had been tampered with by calculating whether the Hamming distance between the CFI labels had the set value. However, these methods usually design CFI labels through static analysis. Therefore, it is difficult for these methods to determine whether the jump directions of branches have been tampered with during program execution.

After preliminary research, we believe that the existing hardware protection methods for conditional branches mainly have three shortcomings:The existing defense methods against BPA make it difficult to distinguish between spy processes and normal processes when applied to lightweight embedded processors;The existing branch information confidentiality protection methods based on PUF make it difficult to resist machine learning attacks;The existing CFI protection methods do not consider whether the jump directions of branches are correct.

To solve these shortcomings, this article proposes a novel hardware security protection method for conditional branches of embedded systems, which can well balance the protection capability, performance overhead, and resource overhead. The main contributions of this article are as follows:The branch information protection mechanism based on hardware locking is proposed. To solve the problem of the existing protection methods against BPA are not suitable for lightweight embedded processors; this mechanism counts the number of updates to the BTB every 80 clock cycles, and if the locking threshold is exceeded (set to 12 in this article), it prevents new data from being written to the BTB until the number of updates is below the unlocking threshold (set to six in this article);The branch information protection mechanism based on dynamic isolation is proposed. To solve the problem of existing APUF-based protection methods being difficult to resist ML attacks, this mechanism incorporates the advantages of double APUFs (DAPUFs) and Feed-Forward APUFs (FF-APUFs) to design a hybrid APUF circuit, which can provide the keys required to encrypt and decrypt branch information. This circuit has four signal paths (each path passes through 32 switch units and has four feed-forward structures), which enhances the randomness between the output response and the input challenge of this circuit to defend against ML attacks;The control flow integrity protection mechanism based on branch labels is proposed. To solve the problem that existing CFI protection methods are difficult to detect whether the jump directions of branches have been tampered with, this mechanism calculates whether the jump conditions of branches are met during program execution to monitor whether the jump directions are correct. This mechanism also monitors whether the instruction codes and the jump addresses have been tampered with, thus protecting the CFI more comprehensively.

The rest of this article is shown as follows. Section 2 describes the related work. Section 3 presents the threat model. Section 4 provides a detailed description of the protection method proposed in this article. Section 5 presents the experimental results. Section 6 discusses the differences between the proposed method and existing methods, analyzes the limitations of the proposed method, and proposes future research directions. Section 7 concludes the main content of this article.

## 2. Related Work

This section describes the current state of research related to conditional branch security protection methods, including preventing obtaining the execution status of branches, encrypting jump information, and monitoring the integrity of jump information.

### 2.1. Prevent Obtaining the Execution Status of Branches

Attackers can exploit the vulnerabilities of BPU (such as Spectre [2]) to implement BPA [20] or BranchScope [23] attacks to obtain the execution status of branches. 

An architectural support scheme against the BPA attacks was proposed [20], which dynamically identified spy processes injected by attackers and prevented them from filling up the BTB. Although the resource overhead of this method is low, it is difficult to balance the protection capability and performance overhead due to the simple mechanism of identifying spy processes. Conditional Speculation [24] dynamically identified dangerous access instructions by implementing different filtering mechanisms but had a high-performance overhead for some test programs. BRB [25] prevented malicious data injected by attackers from being used by assigning separate branch history tables (BHTs) to different programs but had a high-resource overhead and could not provide effective protection for the shared parts.

Therefore, in order to prevent attackers from injecting spy processes, analyzing side channel information such as BTB update time differences, and ultimately determining which branches of the program have been executed, it is necessary to design appropriate hardware protection mechanisms to accurately identify spy processes injected by attackers and prevent them from fully obtaining the execution status of the branch.

### 2.2. Encrypt the Branch Information

The information stored in the BPU is usually stored in plaintext, which can be shared between different threads, posing a serious risk of information leakage.

Zhao et al. [17] cryptographically protected the content and index of the BPU by using randomized lightweight processing to prevent attackers from accessing critical jump information. However, they used a simple hardware random number generator, which resulted in insufficient security for the ciphertext. Researchers in our lab proposed an efficient cryptographic accelerator that could protect the dynamic data security of embedded systems [22]. After that, they proposed two post-quantum cryptographic algorithms that could improve higher levels of cryptographic protection for critical data [11,12]. However, although their hardware circuits have been optimized, the resource overhead and the time of encryption and decryptions are still unacceptable for the BPU.

The PUF can generate random numbers by using uncontrollable process deviations introduced during the manufacturing process, and these random numbers can be used to encrypt critical branch information. HCIC [19] used PUF to encrypt the return addresses and jump addresses, which could resist ROP and JOP attacks. However, it did not describe the PUF circuit structure in detail. The PUFs are divided into weak PUFs and strong PUFs according to their ability to generate challenge–response pairs (CRPs) [26]. Strong PUFs can generate an exponential number of CRPs, mainly including Arbiter PUF (APUF) [27], Ring Oscillator PUF (RO-PUF) [28], etc. The disadvantage of APUF is its vulnerability to modeling attacks based on machine learning. To counteract modeling attacks, researchers proposed structures such as Feed-Forward APUF (FF-APUF) [29] and Heterogeneous APUF [30]. However, it was demonstrated that traditional APUFs and their variant structures had been predicted successfully with a rate of up to 95% or more under targeted machine learning attacks [21]. Compared with APUF, RO-PUF is more secure, but the resource overhead is too large for lightweight applications [31].

Therefore, according to the actual application requirements of the embedded system, it is necessary to consider the security, performance, resources, and other indicators and design a suitable hardware circuit structure to realize the confidentiality protection of the branch information.

### 2.3. Monitor the Integrity of Branch Information

The integrity of jump information means that the instruction codes, jump addresses, and jump directions of the conditional branches were not tampered with when the embedded system executed the program.

A fine-grained hardware protection method for instruction code integrity was proposed by Wang et al. [32]. They used the LHash [33] function to tag the instruction codes and generate unique labels. If the value of the label changes, it means that attackers have successfully tampered with the corresponding code. After that, they proposed a security monitoring and fault recovery architecture for runtime program execution [10], which could safeguard the integrity of partial jump instructions by limiting the target address range. However, its performance overhead was more than 9%. A security method was proposed [13], which not only tagged the instruction codes but also generated CFI labels based on jump relationships. To further optimize the performance overhead and resource overhead, the researchers tagged the jump addresses and generated labels [15]. When the embedded system executes the program, if attackers tamper with the instruction codes or the target addresses, the designed security monitoring module will detect the change in the corresponding labels in time. 

Although the above methods can effectively monitor the integrity of instruction codes and jump addresses, they cannot determine whether the jump directions have been tampered with. Therefore, a more secure and effective hardware monitoring architecture needs to be designed to achieve the fine-grained security protection of conditional branches with a lower-performance overhead and resource overhead.

## 3. Threat Model

This section describes the embedded processor used in this article, the structure of the BPU, and the security threats to BPU.

### 3.1. Xuantie E906 Processor

The conditional branch instructions supported by the E906 are shown in Figure 1 and include BEQ, BNE, BLT, BLTU, BGE, BGEU, C.BEQZ, and C.BNEZ. To obtain the internal details of the embedded processor core, we selected the Xuantie E906 Processor (T-Head Semiconductor Co., Ltd., Hangzhou, China) [34] for research, which is an open-source RISC-V processor. The E906 uses a five-stage pipeline for integer operation, which includes IF (Instruction Fetch), ID (Instruction Decode), EX (Instruction Execution), MEM (Memory access), and WB (Write Back). In this article, we focus on the branch prediction in the IF stage and the instruction monitoring in the ID stage.

### 3.2. Branch Prediction Unit

The structure of the BPU used in E906 is shown in Figure 1. BPU is located in the Instruction Fetch Unit (IFU) and consists of a Branch History Table (BHT) and Branch Target Buffer (BTB). BHT is used to predict the jump directions of conditional branches. The Branch History Register (BHR) is used to record the prediction trajectory and execution trajectory of the previous branches. BHT consists of eight two-bit counters to predict whether a branch will jump or not. BTB is used to predict the jump addresses and contains 16 entries. Each entry contains Valid (indicates whether this entry is valid), Tag (is the lower 16 bits of Branch_Address and is used for indexing), and Target (is the lower 16 bits of Jump_Address). At first, IFU will determine whether the current instruction is a branch based on its opcode. If it is a conditional branch, Branch_Address and BHR_Out are used as the index to find the corresponding two-bit counter and predict whether this branch will be taken or not. Meanwhile, the Branch_Address will be compared with the Tag for all BTB entries. If the Tag of a certain entry matches successfully, the Target of this entry is extracted as BTB_Out. Finally, BPU_Out (which is the address of the next instruction) will be determined based on BHT_Out. If the branch should be taken, BPU_Out is Jump_Address. If the branch should not be taken, BPU_Out is Branch_Address + 0x4.

### 3.3. Security Threats

Figure 1 illustrates the main security threats faced by BPUs. Malicious attackers tamper with or obtain critical jump information (instruction codes, jump addresses, and jump directions) through the vulnerabilities in the BPU. 

Typical tampering attacks include code injection attacks (CIAs) [35], code reuse attacks (CRAs) [36], etc. Attackers inject malicious codes into the embedded systems or use source programs to construct gadgets and successfully tamper with instruction code and critical registers (such as Program Counter, General Purpose Registers, etc.). These malicious actions will seriously affect the data integrity and control flow integrity of the program and cause security incidents. Side channel attacks happen when attackers exploit the side channel vulnerabilities of the BPU to obtain the jump directions and jump addresses of the critical branches. BPA [20] and BranchScope [23] are time-based side-channel attacks. BPA infers whether a critical branch instruction is executed by the difference in update time between the spy processes and the normal processes in the BTB. BranchScope uses a similar principle to attack BHTs by analyzing time differences to determine the execution of critical branches and obtain jump addresses.

Researchers have proposed various security protection methods to address the above security threats, but either the protection scope is insufficient, or the overhead of performance and resources is too high for lightweight embedded systems. Therefore, it is necessary to investigate a novel protection method that can address the major security threats faced by conditional branches with lower performance overhead and resource overhead.

## 4. The Hardware Security Protection Method for Conditional Branches

This article proposes a hardware security method to defend against tamper-type attacks against conditional branches. Figure 2 shows the overall structure of the proposed method. During program execution, the designed security module will dynamically calculate the number of BTB_Updates every 80 clock cycles. If it exceeds the locking threshold (set to 12 in this article), BTB will be locked to prevent attackers from filling up BTB through spy processes and obtaining the execution status of branches by analyzing the time difference in whether BTB is updated. Meanwhile, the designed confidentiality protection module will encrypt the branch information inside BHT and BTB.

If the above protection mechanism fails and attackers successfully hijack the control flow of the program, the designed security monitoring module (SMU) will detect these attacks in a timely manner by comparing the labels of the branch instruction and initiating the processor reset to re-execute the program. When the program is offline, the reference information of the program will be extracted, including Address_INT_ (the lower 16 bits of Branch_Address), Static_Label_Target1_ (the label when the branch will be executed), and Static_Label_Target2_ (the label when the branch will not be executed).

When the binary file of the source program is loaded onto the Flash or DDR, all reference information will be stored in the Integrity Parameter Memory in a secure loading manner. When the program is running, SMU extracts the binary code and address from the ID stage. SMU identifies whether the current extracted instruction is a conditional branch through the opcode shown in Figure 1. As for conditional branches, SMU calculates the Dynamic_Label and compares it with Static_Label_Target1_ or Static_Label_Target2_. If Dynamic_Label is equal to Static_Label_Target1_ or Static_Label_Target2_ and matches the jump direction and address of this branch, it indicates that the control flow is normal. If it is not equal or does not match the jump direction and address of this branch, it indicates that the control flow has been tampered with, and the processor needs to perform the reset operation. By re-executing the program, the embedded system will be restored to normal.

The following text provides a detailed introduction to the branch information protection mechanism based on hardware locking, the branch information protection mechanism based on dynamic isolation, and the control flow integrity protection mechanism based on branch labels.

### 4.1. The Branch Information Protection Mechanism Based on Hardware Locking

In this section, the proposed branch information protection mechanism based on hardware locking is introduced. For BPA attacks [20], the spy processes contain a large number of branch instructions far exceeding those of normal processes, which are used to fill the BTB. Therefore, this mechanism counts the number of BTB_Update every 80 clock cycles, and if the locking threshold is exceeded (set to 12 in this article), it is considered that there is a spy process, and the 1-bit lock field of the BTB will be set to 1, which will prevent any process from modifying the BTB. Until the number of BTB_Updates is below the unlocking threshold (set to 6 in this article), new data can be written to the BTB. In this way, the mechanism can prevent attackers from filling up the BTB and obtaining the execution status of branches.

#### 4.1.1. Identify Characteristics of Attack Behaviors

Figure 3 shows an example of the BTB update process. When the program starts running, IFU extracts the instruction and determines the instruction type based on the opcode. As shown in Figure 3a, BPU predicts whether a branch will jump. BPU will extract the PC value (Branch_Address) of this branch and predict its Jump_Address. It will retrieve whether there is a Tag field of a certain BTB entry, which is equal to the lower 16 bits of the branch address. If equal, the next instruction address to be extracted (Next_PC) is the branch’s Jump_Address of this branch. Otherwise, Next_PC is the Branch_Address + 0x4, meaning that the next instruction will be extracted in order. IDU will determine whether the prediction of BPU is correct based on the actual situation. If the prediction is not correct, it is necessary to update the BTB by writing new data to it. 

An example of writing new data for BTB is shown in Figure 3b. The instruction code is 0x 01d78963, and its PC is 0x 6a2. Based on Figure 1, it can be seen that the opcode of this instruction is 1100011, meaning that it is a branch instruction, and the immediate number of this branch is 0000 0000 1001. The offset of this branch can be calculated by shifting the immediate number one bit to the left and expanding it to 32 bits, which is 0x 12. 

Equation (1) illustrates how the Jump_Address of a branch is calculated.
(1)$$\mathrm{Jump}\_\mathrm{Address}=\left\{\begin{array}{c} Branch\_Address+(\mathrm{Imm}\_\mathrm{Num}<<1),\mathrm{meets}\;\mathrm{conditions}\\ Branch\_Address+0x4,\;\mathrm{not}\;\mathrm{meets}\;\mathrm{conditions} \end{array}\right.$$

As shown in Figure 3b, if the jump condition of this branch is met, its Jump_Address is 0x 6b4, which is Branch_Address (0x 6a2) plus 0x 12. Otherwise, Jump_Address is 0x 6a6, which is Branch_Address plus 0x 4. To save storage space, when writing new data to the BTB, the lower 16 bits of Branch_Address are written to the Tag field of BTB entries, and the lower 16 bits of Jump_Address are written to the Target field. 

Take the BPA attack as an example [20]. Attackers need to build a spy process to fill the BTB as much as possible. Theoretically, the spy process can be identified by the proportion of branches in all instructions or by the occupancy of BTB [20]. However, we found that for lightweight embedded processors (for example, the E906 has only 16 BTB entries), it is difficult for these methods to correctly identify spy processes from normal processes. 

Usually, normal processes do not execute a large number of branch instructions. Therefore, some researchers identify spy processes based on the occupancy rate of BTB [20]. This is very suitable for BTBs with large capacities (e.g., 4096 entries). This is because the average occupancy of the BTB by normal processes is much lower than the average occupancy of the BTB by spy processes (which is usually above 90%). However, the E906 is a lightweight processor with only 16 BTB entries, which means that the BTB occupancy of a normal process may also be high. In addition, the E906 is a sequential single-threaded processor, and compared with the multi-threaded processor, the BTB of the E906 does not record the process, which means that it is impossible to determine whether the current BTB information belongs to a normal process or a spy process. 

The mechanism proposed in this paper identifies the spy process by the number of BTB_Updates every 80 clock cycles. 

E906 has a 5-stage pipeline and 16 BTB entries. Assuming that each instruction of the spy process is a branch, the spy process needs at least 16 instructions to fill the BTB of E906. E906 needs to reach the final stage of the pipeline to decide whether to update the BTB, so the spy process needs at least 80 clock cycles to fill the BTB. Therefore, this article counts the number of BTB_Updates every 80 clock cycles.

Table 1 describes the characteristics of the selected benchmarks. The benchmarks of hello_world and coremark are in the folder of the E906 project (opene906-main\smart_run\tests\cases) [34]. In order to compare with the previous works of our team [10,13,14], we selected eight other benchmarks from Mibench as the test programs. These benchmarks are real-life embedded applications of various scales.

According to Table 1, the proportion of branches in all instructions does not exceed 30%. However, the spy process constructed by attackers may only contain dozens of branches, which can completely fill the BTB of E906 processor without increasing the proportion of branches a lot. Therefore, choosing the proportion of branches in all instructions as the feature for identifying BPA attacks is inappropriate.

Some researchers chose the occupancy rate of BTB as the feature for identifying BPA attacks [20]. The processor they selected had 4096 BTB entries. When this processor executed test programs from Mibench, the highest occupancy rate of BTB did not exceed 35%. However, as shown in Table 1, even if the E906 executes the normal process, the occupancy rate of BTB can be high, even up to 100%. Therefore, for lightweight processors, it is difficult to distinguish between normal processes and spy processes based on the occupancy rate of BTB.

While complex normal processes can also fill the BTB of E906, they will not be written to the BTB at a high frequency in a short period of time. However, the spy process will concentrate on executing a large number of branches to fill the BTB. Therefore, the proposed mechanism chooses the number of BTB_Updates every 80 clock cycles to identify the BPA attacks. As shown in Table 1, the max number of BTB_Update every 80 clock cycles is 8. If the number of BTB_Updates is higher than the locking threshold (set to 12 in this article), it indicates the existence of spy processes and requires protective measures. 

#### 4.1.2. BTB Based on Hardware Locking

Figure 4 shows the principle of the hardware locking mechanism against BPA attacks. As shown in Figure 4a, to execute BPA attacks, attackers first need to build a spy process that contains a large number of branches and use that process to fill the BTB. Afterward, when the embedded system runs a normal process, the prediction will fail due to the fact that the information inside BTB comprises all attacker-constructed data, and the information of branches (including Branch_Address and Jump_Address) is written for the BTB. When the spy process fills up BTB again, the update time of each BTB entry is different. When the embedded system executes the normal process previously, if a branch of the normal process needs to be executed, the information of this branch will be written to the corresponding BTB entry. When executing the spy process again, due to prediction failure, it is necessary to update these BTB entries. However, if a branch of the normal process does not need to be executed, its corresponding BTB entries are still the same as the first time the spy process filled the BTB. Therefore, when the spy process fills the BTB for the second time, it does not need to update these BTB entries. By analyzing the difference in update times of different BTB entries, attackers can successfully predict the branch execution of the normal process. 

The hardware locking protection mechanism proposed in this article adds a 1-bit lock field to each entry of the original BTB. As shown in Figure 4b, if the number of BTB_Update per 80 clock cycles exceeds the locking threshold (Threshold_Lock_), it is considered that there is a spy process, and the lock field of all BTB entries will be set to 1. For a period of time thereafter, no process can write new data to BTB anymore. Only when the number of BTB_Updates per 80 clock cycles is below the unlocking threshold (Threshold_unLock_), the lock field is set to 0, meaning that the BTB can be updated normally. Due to this protection mechanism, attackers will only be able to populate a portion of the BTB entry by performing BPA attacks, resulting in a much lower success rate in predicting whether a branch will execute or not.

Table 2 demonstrates the impact of Threshold_Lock_ and Threshold_unLock_ on the protection effectiveness and performance overhead. In this article, we evaluate the protection effectiveness of the proposed mechanism by the success rate of BPA attacks and evaluate the performance overhead by the growth rate of cycles per instruction (CPI). The attack is considered successful if attackers can accurately predict whether a branch will jump or not. Assuming that the test program executes 100 branches during runtime and attackers successfully predict whether 70 branches will jump or not, the attack success rate is 70%. As can be seen from Table 1, coremark has the maximum number of BTB_Updates per 80 clock cycles, which is 8, so it is chosen as the test program. 

In order to find the most suitable Threshold_Lock_, we first test the impact of different Threshold_Lock_ on the protection effectiveness and performance overhead. The Threshold_unLock_ is set to 1 at this time. As shown in Table 2, the lower the Threshold_Lock_, the better protection effectiveness, but the higher the performance overhead. If Threshold_Lock_ is very low, the spy process starts to fill up the BTB, and the number of BTB_Update will exceed the Threshold_Lock_, resulting in a period of time afterward when the BTB is locked and cannot be updated. At this point, the performance overhead increases dramatically as the BTB is locked frequently. Moreover, if the Threshold_Lock_ is very low, the problem of difficulty in distinguishing normal processes from spy processes can also occur. Therefore, considering the protection effectiveness and performance overhead, the Threshold_Lock_ is set to 12 in this article.

To find the most suitable Threshold_unLock_, we set Threshold_Lock_ to 12 and test the impact of a different Threshold_unLock_ on the protection effectiveness and performance overhead. As shown in Table 2, the lower the Threshold_unLock_, the better protection effectiveness, but the higher the performance overhead. If the Threshold_unLock_ is high and close to the Threshold_Lock_, the BTB will be unlocked quickly, causing a small performance overhead. However, if a spy process exists at this point, BPA attacks can still be successful. Therefore, considering the protection effectiveness and performance overhead, the Threshold_unLock_ is set to 6 in this article.

### 4.2. The Branch Information Protection Mechanism Based on Dynamic Isolation

The protection mechanism based on hardware locking can prevent attackers from tampering with the internal data of BTB through spy processes, thereby obtaining critical branch information of normal processes. However, the jump direction and jump address of branches are often stored in plaintext in BHT and BTB, which attackers can obtain through other means. Therefore, there is still a certain risk of information leakage in BPU.

Most existing research adopts methods such as logical isolation or physical isolation to protect the key information within BPUs [17]. Logical isolation is to prevent other units from sharing branch predictor data, and a common implementation is to refresh the branch predictor when process switching or permission changes occur. However, refresh-based logical isolation schemes often have a significant impact on performance and cannot resist side-channel attacks based on competition. Physical isolation is the allocation of separate branch predictor tables for different threads and privilege levels, which can defend against content reuse-based attacks and competition-based side-channel attacks, but the hardware resource overhead is too high. The PUF-based dynamic isolation mechanism for branch information proposed in this article can generate proprietary random numbers through the designed hybrid PUF circuit, which can be used as keys to encrypt the key branch information and their index, preventing obtain and analysis by attackers and achieving dynamic isolation of jump directions and jump addresses.

#### 4.2.1. BTB and BHT Based on Dynamic Isolation

Figure 5 shows the dynamic isolation mechanism proposed in this article. This mechanism encrypts the data written to BHT and BTB during the execution of embedded system programs and only decrypts data when they are needed. Since the critical data about branch addresses and jump addresses within the BHT and BTB are encrypted, it is difficult for attackers to obtain valid information in a short period of time, even if they can successfully steal these data. Therefore, this mechanism enables dynamic isolation of the branch information during program execution and prevents attackers from stealing these data.

As shown in Figure 5a, when the program is running, this mechanism concatenates the higher 16 bits of the Branch_Address with the output of the branch history register (BHR_Out) and generates a digest through the LHash module as the index of the corresponding 2-bit counter. The designed confidentiality protection module will generate two keys (a 16-bit Key for Branch_Address and an 8-bit Key for Index), one for encrypting the lower 16 bits of Branch_Address stored in BHR and the other for encrypting the digest used as the index. In this way, attackers will find it difficult to access BHT and obtain the jump directions of branches.

LHash is a hash function that can be used to generate index values for data [33]. Compared to other hash functions, LHash can provide the required security with minimal resource overhead, making it suitable for lightweight embedded processors [15]. Considering security and area consumption, the LHash module designed in this article uses the 128-bit internal permutation structure and outputs the 96-bit binary number. Due to the BHT size of the E906 processor being 512 (2^8^) × 16 and having eight 2-bit counters, the length of the index is 8 bits, which randomly selects 8 bits from the 96-bit output of LHash as the index for the 2-bit counter. Therefore, the length of the Key for Index is 8 bits. Due to BHR storing the lower 16 bits of branch addresses, the length of the Key for Branch_Address is 16 bits.

As shown in Figure 5b, when it is necessary to update the internal data of BTB, the designed confidentiality protection module will generate two keys (a 16-bit Key for Branch_Address and a 16-bit Key for Jump_Address). The lower 16 bits of Jump_Address will be encrypted and stored in the Target field, and the lower 16 bits of Branch_Address will be encrypted and stored in the Tag field. When it is necessary to predict the jump address of a branch, this module will generate two keys for that branch again. The Key for Branch_Address will XOR with the lower 16 bits of Branch_Address and compare the calculated value with the Tag of each BTB entry. Only when the values of a certain entry match successfully, will BTB output the Target of this entry. The plaintext of BTB_Out can only be obtained after the Target is XOR with the Key for Jump_Address. In this way, attackers will find it difficult to access BTB entries and obtain the branch information.

Our team has proposed various security algorithms and their hardware circuits that can be used for information encryption [11,12]. However, although these algorithms have extremely high security and can defend against quantum attacks, they have high resource overhead, long encryption and decryption times, and are not suitable for BHT and BTB. Our team has also designed the hardware accelerator based on traditional cryptographic algorithms such as AES-GCM [22]. However, this module is more suitable for confidentiality protection applications with large amounts of data and has high resource overhead. Therefore, it is not suitable as well. Taking into account factors such as security, resources, and performance, the confidentiality protection module of this article is based on the designed hybrid PUF circuit and efficiently generates multiple keys required for the dynamic isolation mechanism.

Due to the different keys for different branches, attackers are unable to restore correct branch jump information and, therefore, cannot correctly perceive whether there is competition with other processes, nor can they reuse previous historical information, fundamentally resisting competition-based and reuse-based attacks. The proposed dynamic isolation mechanism does not change the original control logic for updates and queries; it only encrypts the jump information and their index with a low overhead of resources and performance. This mechanism minimizes changes to the existing branch predictor architecture and only requires a small amount of logic to achieve dynamic isolation.

#### 4.2.2. The Confidentiality Protection Module

Figure 6 shows the main components of the confidentiality protection module. Due to constraints such as hardware resources, cost, and computing power, traditional encryption techniques are difficult to widely apply in lightweight embedded systems. Physical Unclonable Function (PUF), as a physical security primitive, utilizes the uncontrollable random process deviations introduced during chip manufacturing to generate unique secret keys [31]. A set of n-bit Challenges is applied to the PUF circuit, corresponding to the generation of m-bit Response, which is called CRP. According to the different abilities of PUFs to generate CRPs, they are divided into weak PUFs and strong PUFs [26]. Strong PUFs can generate an exponential number of CRPs suitable for low-cost encryption and decryption, mainly including APUF based on arbitrators, RO-PUF based on ring oscillators, and current mirror PUF [28]. RO-PUF and current mirror PUF are susceptible to environmental noise interference and have high hardware resource costs, making them unsuitable for lightweight encryption and decryption applications. Therefore, this article chooses the APUF circuit to implement the confidentiality protection module.

As shown in Figure 6a, the confidentiality protection module consists of three main parts. The pulse generator is used to generate the pulse signal as an input signal to the hybrid APUF circuit. The challenge generator is used to generate a 32-bit challenge signal (C_1_, C_2_, ……, C_32_). Through the 32-bit linear feedback shift register (LFSR), the challenge generator expands the input 32-bit Branch_Address or Jump_Address to the maximum of 2^32^–1 pseudo-random 32-bit challenge signals, which will loop between 2^32^-1 different states. In this way, the randomness of the generated keys is enhanced to defend against ML attacks. The hybrid APUF circuit outputs a key based on the input pulse and challenge signals. The hybrid APUF circuit has multiple paths. This circuit controls which path the pulse signal passes through based on the challenge signals and determines whether the output response is 1 or 0 based on the arrival time of the pulse signal. For the proposed hybrid APUF circuit, each 32-bit challenge signal can generate a 1-bit response. Therefore, to generate the 16-bit Key for Branch_Address, the initial input signal is the 32-bit Branch_Address, and then sixteen challenge signals are randomly selected from the output sequence of the LFSR. After 16 clock cycles, the required key can be generated. Similarly, the initial input signal is the 32-bit Jump_ Address, and after 16 clock cycles, the 16-bit Key for Jump_Address can be obtained. As for the 8-bit Key for Index, the initial input signal of the challenge generator is the 32-bit Branch_Address, and the eight challenge signals are randomly selected from the pseudo-random sequence of the LFSR.

The circuit of the proposed hybrid APUF is shown in Figure 6b. This circuit incorporates the advantages of double APUFs (DAPUFs) and Feed-Forward APUFs (FF-APUFs), which enhances the randomness between the output response and the input challenge to defend against ML attacks. This circuit has four signal paths, and each path passes through 32 switch units and has four feed-forward structures. Since the E906 is a 32-bit processor, the Branch_Address and Jump_Address are 32 bits; thus, the challenge signals are 32 bits. One bit of the challenge signal controls one switch unit, so each path of the hybrid APUF circuit needs to pass through 32 switch units.

The traditional APUF circuit consists of a signal delay path cascaded with n switch units and an arbiter [27], and a switch unit consists of two parallel multiplexers (MUX). When the challenge signal Ci is 0, the signal path of this switch unit is the parallel path, and when Ci is 1, the signal path is the cross path. Therefore, the transmission path of the signal can be changed by controlling the challenge signal C, which affects the signal delay difference and generates an unpredictable response. The arbiter is a NAND latch. When the pulse signal is applied to the input of the APUF circuit and after being transmitted through two symmetrical delay paths (Path 0 and Path 1), the arbiter arbitrates the output signals of Path 0 and Path 1. If the signal of Path 0 arrives at the arbiter before the signal of Path 1, the output of this PUF circuit is 1; otherwise, it is 0.

The total delay deviation of the signal reaching the arbitrator is the accumulation of delay deviations when it passes through each switch unit, as shown in Equation (2):
(2)$$\Delta =(\prod_{i=1}^{n}\left(1-2c_{i}\right))•{\left(\omega_{i}\right)}^{T}$$

c_i_ is a sub-element of the Challenge vector C, and ω_i_ is a constant vector containing delay parameters for each switch unit, as shown in Equation (3):
(3)$$\begin{array}{c} \omega_{1}=\alpha_{1}\\ \omega_{i}=\alpha_{i}+\beta_{\left(i-1\right)}\\ \omega_{n+1}=\beta_{n} \end{array},2\leq i\leq n$$

The calculation formula of α_i_ and β_i_ are shown in Equation (4):
(4)$$\begin{array}{c} \alpha_{i}=\frac{x_{i}-y_{i}-u_{i}+v_{i}}{2}\\ \beta_{i}=\frac{x_{i}-y_{i}+u_{i}-v_{i}}{2} \end{array}$$

x_i_ represents the delay parameter of the signal passing parallel through the delay Path 0 when the challenge signal C_i_ is 0, and y_i_ represents the delay parameter of the signal passing parallel through the delay Path 1 when C_i_ is 0. u_i_ represents the delay parameter of the signal crossing through the delay path (from Path 0 to Path 1) when C_i_ is 1, and v_i_ represents the delay parameter of the signal crossing through the delay path (from Path 1 to Path 0) when C_i_ is 1.
(5)$$R=\frac{\mathrm{sgn}(\Delta )+1}{2}$$

The relationship between the response R of the APUF circuit and the total delay deviation ∆ is shown in Equation (5). The value range of the sgn function is {−1, 1}. When ∆ is greater than 0, R is 1; otherwise, R is 0.

Due to the inherent correlation between the output response and input challenge of APUF, it is susceptible to ML attacks [21]. Attackers can collect a number of CRPs and use ML algorithms to build a mathematical model to predict a response to an arbitrary challenge with high accuracy. At present, researchers mainly resist ML attacks by modifying APUF circuits to enhance the nonlinearity of circuit models [29]. Some researchers proposed FF-APUF, which introduced a feed-forward loop in APUF and used the decision results of the intermediate stage as the excitation for the subsequent stage switch unit [30]. Some researchers proposed the DAPUF, which configured the signals sent to the arbiter for judgment according to the principle of cross-exchange, enhancing the ability of APUF to resist ML attacks [37]. However, other researchers pointed out that these structures had a success rate of over 95% predicted under ML attacks [21]. Therefore, this article combines the advantages of FF-APUF and DAPUF circuits to deepen the nonlinearity of the hybrid APUF circuit structure and improve the ability to resist ML attacks.

The proposed hybrid PUF circuit is shown in Figure 6b. Add a D trigger at the beginning of each delay path (Path 0, Path 1, Path 0′, and Path 1′) to optimize the routing delay of the PUF circuit. These D triggers are driven by the same clock. Only when the rising edge of the clock arrives, will the trigger output the signal, ensuring that the time for the pulse signal input to each delay path is consistent.

In order to disrupt the mapping relationships between challenges and responses and reduce the accuracy of key inference based on ML attacks, a mirrored APUF with the same structure as the original APUF is designed, ensuring that the original APUF is parallel and symmetrical with the mirrored APUF. The outputs of the original APUF and the mirror APUF will be obfuscated by XOR to obtain the 1-bit output response. Due to the comparison signals of the Arbiter 5 and Arbiter 5′, both coming from the same type of delay paths, the asymmetry of the APUF signal delay path will be effectively compensated in the hardware implementation, which can improve the ability of the PUF circuit to resist ML attacks.

To expand the selection range of signal delay paths, enhance the randomness of the PUF circuit, and achieve maximum utilization of switch units, the lower delay path and upper delay path of the symmetrically set switch units output are crossed with each other, achieving mutual inversion of input excitation. For example, Path 1 is no longer the parallel path from Switch Unit 7 to Switch Unit 8 but crosses to Switch Unit 8′. Similarly, Path 0′ will move from Switch Unit 7′ to Switch Unit 8. Unlike traditional APUF circuits, the pulse signals of hybrid PUF circuits can be transmitted through the cross path on the mirror-symmetric APUF delay path, which can effectively improve the randomness of the circuit output response and resist ML attacks.
(6)$$\begin{array}{l} Num_{FF}=\frac{n}{Num_{SU}+1}\\ Num_{SU}=2^{j}-1,j=1,2,3,\ldots \ldots ,\log_{2}(n)-1 \end{array}$$

The feed-forward loop of the PUF circuit is shown in Figure 6b, and its quantity has a significant impact on the stability and ML attack resistance of the PUF circuit. In order to determine the optimal feed-forward loop, we conducted experiments, and the test results are shown in Table 3. The relationship between the number of feed-forward circuits Num_FF_ (only consider the original APUF) and the number of interval switch units Num_SU_ is shown in Equation (6). n is the total number of switch units. Each path of the hybrid PUF circuit has a total of 32 switch units, so the selectable range of the Num_SU_ is {1, 3, 7, 15} and the selectable range of the Num_FF_ is {2, 4, 8, 16}.

Stability is measured by the average intra-Hamming distance (HD), and the calculation formula is shown in Equation (7):
(7)$$\begin{array}{l} Stability=1-\frac{1}{T}\sum_{t=1}^{T}\frac{HD(R,R_{t})}{L}\times 100\%\\ HD(R,R_{t})=\sum_{k=0}^{L-1}R[k]⊕R_{t}[k] \end{array}$$

R represents the output response of the hybrid PUF circuit at 30 degrees centigrade and standard voltage, and Rt represents the output response of the t-th time under different environmental conditions. T represents the total number of environmental conditions, which is the total number of times the circuit response is measured. L is the length of the output response. HD(R, R_t_) is the Hamming distance between the responses R and R_t_. This article will implement the designed hybrid PUF circuit on FPGA and test its stability in the temperature range of 25 to 75 degrees centigrade with a step size of 10 degrees centigrade. Each experiment will be repeated 10 times, and the average stability under multiple environmental conditions will be calculated. Based on the test results in Table 3, taking into account the stability, resistance to ML attacks, and hardware resource overhead, Num_FF_ and Num_SU_ selected in this article are 4 and 7, and the structure of the hybrid PUF circuit is shown in Figure 6b.

### 4.3. The Control Flow Integrity Protection Mechanism Based on Branch Labels

If the protection mechanism proposed above fails, attackers can manipulate the jump directions or addresses of branches and hijack the control flow of the program through CIAs [35], CRAs [36], etc. Although researchers have proposed various fine-grained control flow integrity (CFI) monitoring methods [10,13,14,15,32] that can detect tampering attacks on instruction addresses, instruction code, and jump addresses in a timely manner, these methods cannot determine whether the jump direction of branches during program execution has been tampered with. This article proposes the CFI monitoring mechanism based on branch labels, which integrates the jump conditions of branches into the label calculation process. By comparing whether the dynamic and static labels are equal and calculating whether the jump conditions of branches are correct, the proposed mechanism can protect the CFI more comprehensively.

#### 4.3.1. Extract the Reference Information

Figure 7a illustrates the main process of extracting reference information. Firstly, compile and link the source code to generate the binary code. Secondly, disassemble the binary code to obtain the disassembly text. Afterward, perform static analysis on the text and divide instructions into several basic blocks (BBs). BB is a collection of sequentially executed instructions and only contains one jump instruction. Analyze the information of BBs, including the BB Address (the address of the first instruction in the BB), the binary code of all instructions in the BB, and the jump address of the jump instruction in the BB. Based on this information, extract the corresponding reference information.

Figure 7b shows an example of extracting reference information. According to the BB partitioning rules, this part of the disassembly text can be divided into three BBs, which are BB_i−1_, BB_i_, and BB_i+1_. BB_i−1_ and BB_i+1_ only have one instruction, which is a BEQ-type instruction. BB_i_ has four instructions, and the last is a BLTU-type instruction. Taking BB_i_ as an example, its BB Address is 0x 6a6. The jump condition of BB_i_ is to determine whether the unsigned number of register a5 is less than the unsigned number of register t0. If the jump condition is met, the jump address of BB_i_ is 0x 6a2, which is the address of BB_i−1_. Otherwise, the jump address is 0x 6b4, which is the address of BB_i+1_.

Further, process the BB information to obtain the reference information. As shown in Figure 7b, take the lower 16 bits of the BB Address as Address_BB_, which can be used for indexing the reference information of this BB. Using the LHash algorithm [33], map the binary codes of all instructions within the BB to a 16-bit digest. If attackers tamper with any instruction code of this BB, the corresponding digest will also change. Therefore, the digest represents the integrity of the instruction code within BBs. Calculate the Hamming distance (HD) between the jump address and the branch instruction address (Branch_Address) of the BB, and XOR the result with the digest to obtain 16-bit Static_Lable_Target1_ and Static_Lable_Target2_. If attackers hijack the control flow and alter the execution order of instructions, the calculated HD will also change. Therefore, HD can represent the CFI of BBs. If the only jump instruction within the BB is not a conditional branch but an unconditional jump, the reference information for that BB only has valid Address_BB_ and Static_Lable_Target1_, while Static_Lable_Target2_ is null.

This article considers both instruction code integrity and control flow integrity when designing static labels. When the program is running, the designed SMU analyzes the decoded instructions and divides these instructions into several BBs. Referring to the static analysis process, SMU calculates the digest and the Hamming distance and XOR to obtain the Dynamic_Label of BBs. Based on the Address_BB_, SMU can find the corresponding static labels. For the BBs containing unconditional jumps, check whether Dynamic_Labels are equal to Static_Lable_Target1_. For the BBs containing conditional branches, check whether Dynamic_Label is equal to Static_Lable_Target1_ or Static_Lable_Target2_. If the dynamic and static labels are not equal, it is considered that instruction codes or program control flow have to be tampered with.

#### 4.3.2. Monitor the Control Flow Integrity

Although SMU can detect tampering with instruction code or control flow in a timely manner by comparing dynamic and static labels, it is sometimes difficult to determine whether the jump directions of conditional branches have been tampered with. It is difficult to accurately determine whether the jump conditions of conditional branches are met during program execution. Although it is possible to record the execution path of BBs and the execution status of conditional branches by software simulators, the required amount of data is too large and not suitable for CFI monitoring in lightweight embedded processors. Therefore, the proposed CFI protection mechanism will incorporate the jump conditions into the Dynamic_Label calculation. 

Figure 8
shows the CFI monitoring process for branches. When the embedded system starts running the program, SMU extracts instruction information from the e906 processor pipeline, including 32-bit binary code from the instruction register (IR) and 32-bit instruction address from the program counter (PC). Based on the opcode, the instruction type can be identified, and these instructions can be divided into several BBs. Through dynamic analysis, the information of each BB is extracted, including the address of the BB (Address_BB_), the types of jump instructions included, the jump address, and the digest of all internal instruction codes.

The CFI monitoring process for the BBs, including unconditional jumps, is shown in Figure 8a. If the jump instruction within the BB is not a conditional branch but an unconditional jump, the Dynamic_Label of this BB is first calculated. Meanwhile, based on the address, retrieve the Static_Label_Target1_ of the corresponding BB. Due to the unconditional jump having only one jump address, if the dynamic and static labels are different, it is considered that the integrity of the program instruction code or control flow has been tampered with. The processor reset operation is initiated, and the program is re-executed. If the labels are consistent, continue executing the instructions until the program ends.

The CFI monitoring process for the BBs, including conditional branches, is shown in Figure 8b. If the jump instruction within the BB is a conditional branch, determine the branch type based on the binary code [14:12]. Different types of branches have different jump conditions. The condition for the BEQ-type branch is whether rs1 and rs2 are equal. The condition for the BNE-type branch is whether rs1 and rs2 are not equal. The conditions for the BLT and BLTU-type branches are whether the value of register rs1 is lower than the value of register rs2. The BLT-type branch compares signed numbers, while the BLTU-type branch compares unsigned numbers. The conditions for the BGE and BGEU-type branches are whether the value of rs1 is greater than or equal to the value of rs2. The C.BEQZ-type branch is a compression instruction with a length of only 16 bits, and its jump condition is whether rs1 is equal to zero. The condition for the C.BNEZ-type branch is whether rs1 is not equal to zero. To dynamically determine whether the jump condition is met, SMU calculates the difference (O) between the internal data of rs1 and rs2. As for the BNE-type branch, if O is 0, it indicates that the jump condition is met, and the jump address is the current PC value plus the offset. At this time, it is necessary to compare whether the Dynamic_Label is consistent with Static_Label_Target1_. If O is not 0, it indicates that the jump condition is not met at this time. The Dynamic_Label and Static_Label_Target2_ should be compared. If the dynamic and static labels are the same, the program continues to be executed; otherwise, the CPU reset is initiated.

## 5. Experiments and Results

This article implements the proposed hardware protection method for conditional branches on the FPGA platform. The selected FPGA chip is Xilinx Kinex-5, XC7K325T-2FFG900I (XILINX, San Jose, CA, USA), and the development board is ALINX AX7325. We have implemented an embedded system-on-chip (SoC) on the FPGA based on an open-source RISC-V processor (Xuantie E906 [34]). The clock frequency of SoC is 100 MHz. As shown in Table 1, we have selected ten sets of commonly used programs or algorithms in the industrial control field as benchmarks in the open-source project of E906 [34] and Mibench [10]. 

### 5.1. Security Analysis

There are currently many attacks targeting branch information, such as BPA attacks [20], critical data theft [17], and control flow tampering [15]. Therefore, this article proposes three hardware protection mechanisms for branch information to prevent attackers from completely tampering with BTB, dynamically isolating critical data of BHT and BTB, and monitoring the control flow integrity during program execution. The following text will analyze the security of the proposed mechanism separately.

#### 5.1.1. Security of the Hardware Locking Protection Mechanism

BPA [20] is a side-channel attack against BTB. The attack process is shown in Figure 4a. If attackers can accurately predict whether a branch of benchmarks will jump, then this BPA attack is considered successful. Assuming that the test program executes p branches during runtime, and attackers successfully predict whether q branches will jump, the attack success rate is q ÷ p × 100%.

Table 4 shows whether the hardware locking protection mechanism has an impact on the defense against BPA. Facing the unprotected embedded system, the success rate of BPA attacks is close to 100%. This is because attackers can theoretically inject spy processes before the normal process executes branches and completely tamper with BTB. When a branch of the normal process needs to be executed, due to its information being different from the spy process, BTB needs to be updated. Therefore, attackers can theoretically fully understand the branch execution status of the normal process through the difference in BTB update time. However, if there is a conditional branch in the first few instructions when the test program starts running, this attack may fail because the spy process has not yet fully filled the BTB. Therefore, even without protection, the success rate of attacks is not 100%.

With the proposed hardware locking protection mechanism, if the number of BTB_ Update every 80 cycles caused by spy processes exceeds the Threshold_Lock_, the lock field of BTB entries will be set to 1 to prevent BTB from being completely tampered with. Therefore, attackers will not be able to obtain the execution status of most branches and can only make random guesses, resulting in a decrease in attack success rate to nearly 50%. This indicates that the proposed branch information protection mechanism based on hardware locking can effectively resist side-channel attacks such as BPA.

#### 5.1.2. Security of the Proposed Hybrid APUF

The proposed branch information protection mechanism based on dynamic isolation uses the designed confidentiality protection module to encrypt and decrypt branch addresses, jump addresses, and their indexes. Therefore, the security of this mechanism mainly depends on the cryptographic circuit used, which is the hybrid APUF.

Table 5 shows the security comparison of the multiple APUF circuits. Since the different researchers proposed that APUF circuits used different sizes of challenge, even if the type was the same, the structures of APUF circuits are different, and there are differences in safety. Therefore, this article refers to previous research results [21,27,29,38] and designs the traditional APUF, 2XOR-APUF, DAPUF, and FF-APUF. The challenges of these circuits are consistent with the proposed hybrid APUF, which is 32 bits.
(8)$$Randomness=\frac{1}{n}\sum_{e=1}^{n}R_{e}\times 100\%$$
(9)$$Uniqueness=\frac{2}{k(k-1)}\sum_{f=1}^{k-1}\sum_{g=i+1}^{k}\frac{HD(R_{f},R_{g})}{n}\times 100\%$$

Stability is a measure of whether the APUF circuit can ensure the same response from the same challenge in different environments. The calculation of stability is shown in Equation (7). Ideally, the stability should be 100%. Randomness reflects the distribution of 0 and 1 in the response, and ideally, it should be 50%. The calculation of randomness is shown in Equation (8). Assuming the response has n bits, R_e_ is the response of the e-th bit. Uniqueness reflects the ability of different PUF entities to respond differently to the same incentive, ideally at 50%. The calculation of randomness is shown in Equation (9). k represents the number of APUF entities; HD represents the Hamming distance between the response R_f_ generated by the f-th entity and the response R_g_ generated by the g-th entity. This article implements APUF circuits with the same structure at five different positions inside the FPGA, which can be seen as implementing five different APUF entities.

Referring to previous research [38], this article also selected three typical machine learning (ML) algorithms (CMA-ES, LR, and ANN) to predict the output response of APUF circuits. The covariance matrix adaptation–evolution strategy (CMA-ES) is one of the mainstream attack algorithms for PUF, which can customize fitness functions based on the structure and has high-response prediction accuracy. Logistic regression (LR) is a linear regression model with a simple structure and low computational complexity, which can also be used to predict the response. This article uses a gradient descent algorithm as the optimizer of LR, with a learning rate of 0.01, and softmax is used to activate the LR layer. Artificial Neural Network (ANN) is a complex network composed of a large number of interconnected neurons, which can be used to fit the input and output correlation functions of APUF. The ANN designed in this article has three hidden layers (each layer has 128 nodes and is connected to a dropout layer), uses ReLU for nonlinear transformation, and uses softmax for activating the output layer. The learning rate of this ANN is 0.01, and the optimizer is the momentum algorithm. The training set for the three ML algorithms consists of 100 K CPRs, while the testing set consists of 10 K CPRs.

As shown in Table 5, the hybrid APUF proposed in this paper has stronger resistance to machine learning attacks and better security compared to traditional APUF, 2XOR-APUF, DAPUF, and FF-APUF. The stability, randomness, and uniqueness of the hybrid APUF circuit are also close to ideal values.

#### 5.1.3. Security of the Control Flow Integrity Protection Mechanism

In order to evaluate the security of the proposed CFI protection mechanism, this paper designs an attack circuit inside the processor. This circuit can tamper with signals and registers and simulate attacks on conditional branches and unconditional jumps. Due to the conditional branches having two potential-jump addresses, compared to unconditional jumps, attackers can not only tamper with the instruction codes or jump addresses but also tamper with the jump directions to disrupt program control flow. For each benchmark, the attack circuit will randomly select 1000 attack points to tamper with instruction code, jump addresses, or jump directions.

Table 6 shows the detection rates of the proposed mechanism against different types of control flow tampering attacks. From the data in Table 6, it can be seen that the proposed protection mechanism can effectively detect tampering attacks on conditional branches. This is because if an attacker tampers with the instruction codes of branches, the digest calculated by SMU will undergo significant changes compared to the digest calculated during static analysis, resulting in inconsistent dynamic and static labels. Similarly, if the jump addresses are tampered with, the HD calculated by the SMU will be different from the HD calculated during static analysis, also resulting in inconsistent dynamic and static labels. Moreover, compared to our previous research [13,15], since this article calculates whether the jump condition is met, SMU will choose whether to compare the Dynamic_Label with Static_Label_Target1_ or Static_Label_Target2_ based on the actual jump situation. Therefore, this mechanism can detect tampering with the jump directions. 

However, although this mechanism can effectively detect tampering with the instruction codes of unconditional jumps, the attack detection rate for the jump addresses can only reach over 90%, not 100%. This is because unconditional jumps contain indirect jump-type instructions (JALR). If attackers only tamper with the values of registers involved in the JALR instructions and the tampered jump addresses are valid, SMU will not detect these attacks.

### 5.2. Resource Overhead Assessment

The hardware locking BTB proposed in this article only requires the addition of the 1-bit lock field. The BTB of the E906 processor has 16 entries, so the hardware resource overhead is 1 × 16 = 16 bits. Adding the circuit that calculates the number of BTB_Update within 80 cycles, as well as the circuit that controls whether BTB is locked based on the number of BTB_Update, Threshold_Lock_, and Threshold_Unlock_, the total resource overhead of the proposed mechanism is only 110 bits, which can be almost negligible compared to existing protection methods. For example, the anti-BPA protection method proposed in [20] has a resource overhead of 8 KB.

Table 7 shows the hardware resource overhead of different confidentiality protection circuits. Referring to Refs. [21,27,29,38], we designed different circuits based on traditional APUF, 2XOR-APUF, DAPUF, and FF-APUF. Moreover, except for the APUF structure, these circuits have the same circuit components as the confidentiality protection circuit based on the hybrid APUF proposed in this article, such as the pulse generation module, the challenge generation module, and so on. As shown in Table 7, the proposed confidentiality protection circuit consumes more LUTs and Slices than traditional APUF and FF-APUF. This is because, in order to enhance security, this article adds a mirrored APUF circuit, which can confuse the output response and reduce the prediction accuracy based on machine learning algorithms. Compared with the circuits based on 2XOR-APUF and FF-APUF, the resources consumed by the circuit proposed in this article are basically equivalent.

The proposed CFI mechanism requires additional areas to store reference information. Compared to our previous work [10,13], this article combines instruction code integrity and control flow integrity when calculating the static labels of BBs, reducing the reference information of each BB to only 48 bits in size. The reference information size in Ref. [10] is the same as the program code size, while each BB in Ref. [13] has 64-bit reference information. Therefore, compared to the aforementioned studies, the method proposed in this article incurs less resource overhead caused by reference information.

The SoC based on E906 consumed 34,173 LUTs, 14,426 Slice Registers, and 10,529 Slices without incorporating the method proposed in this article. The three proposed hardware protection mechanisms are implemented on FPGA. Including the hardware locking BTB, confidentiality protection circuit based on hybrid APUF, and security monitoring module based on branch labels designed in this article, the total resource overheads of the proposed methods are 1851 LUTs, 816 Slice Registers, and 613 Slices. Therefore, the proposed method only increased the resource cost of LUTs by approximately 5.4%.

### 5.3. Performance Overhead Assessment

This article chooses the cycles per instruction (CPI) to evaluate the embedded system performance overhead caused by the proposed method. Table 8 shows the CPI changes in embedded systems executing different benchmarks before and after the addition of the proposed method. It can be seen from Table 8 that for each benchmark, the performance overhead is less than 5.5%, and the average value is 4.59%. The performance overhead mainly comes from the time spent calculating dynamic labels and retrieving static labels, when SMU monitors the CFI of programs. For the method proposed in this article, the more jump instructions the program needs to execute, the higher the performance overhead. However, for the lightweight embedded processors, in their typical application scenarios, the executed program is not very complex and does not contain too many jump instructions. Therefore, the performance overhead caused by the proposed method is acceptable for lightweight processors. Compared with existing research on control flow protection, such as references [10] (its performance overhead is about 9.33%) and [13] (its performance overhead is about 6.18%), the performance overhead of the proposed method is acceptable for lightweight embedded systems.

## 6. Discussion

This section compares the proposed method with existing methods, analyzes the limitations of the proposed method, and proposes future research directions.

### 6.1. Comparison with Other Security Protection Methods

Table 9 shows the comparison of this article with other security protection methods in terms of security, resource overhead, and performance overhead. When evaluating security, this article focuses on the ability of different methods to protect branch information, such as preventing attackers from tampering with the BPU, protecting the confidentiality of branch information, and monitoring the CFI of the programs.

Ref. [16] proposed a software protection method (Conditional Speculation) that limited the execution of memory instructions, which could resist attacks based on Spectre vulnerabilities. However, this method cannot prevent attackers from stealing branch information or monitoring the CFI. Moreover, although the average performance overhead of this method is low, for special cases, it may limit the execution of normal memory instructions, resulting in a sharp increase in performance overhead.

Ref. [17] proposed a secure branch predictor that encrypted the contents and indexes of BTB and BHT to resist malicious perception attacks on branches. However, the security of this method mainly depends on the random number required for encryption. Attackers can successfully predict random numbers based on ML algorithms and obtain branch information predicted from this predictor. 

Ref. [20] proposed a hardware method to resist BPA, which calculated whether the BTB occupancy rate exceeded the threshold and determined whether BTB updates were blocked. However, this method did not consider the possibility of normal processes being filled with BTB if the BTB entries were small. Therefore, this method is not suitable for lightweight processors like E906. 

Ref. [19] proposed a hardware-assisted control flow checking method (HCIC) that used PUF to encrypt return addresses and jump addresses, which could resist CRAs. However, Ref. [19] did not discuss the countermeasures of PUF against ML attacks, which posed security risks.

Refs. [10,13] monitored the CFI based on label comparison and promptly detected tampering with instruction codes and jump addresses. However, these methods did not consider whether the jump conditions of branches were met and could not detect attackers tampering with the jump directions.

Compared to the above methods, the protection method proposed in this article can prevent attackers from obtaining the execution status of branches, encrypt the branch information, and detect tampering with instruction codes, jump addresses, and jump directions. As shown in Table 1, for lightweight embedded processors like E906, the occupancy rate of BTB will also be high, even reaching 100%. Therefore, the existing protection methods against BPA [20] are not suitable for lightweight embedded processors. This article proposes the branch information protection mechanism based on hardware locking, which counts the number of BTB_Update every 80 clock cycles, and if the locking threshold (Threshold_Lock_) is exceeded, it prevents new data from being written to the BTB until the number of updates is below the unlocking threshold (Threshold_unLock_). As shown in Table 2, taking into account both protection effectiveness and performance overhead, this article sets the Threshold_Lock_ to 12 and the Threshold_unLock_ to 6. As shown in Table 4, after adding the mechanism proposed in this paper, the attack success rate will be reduced from about 98% to less than 60%. Therefore, this protection mechanism can prevent attackers from obtaining the execution status of branches via BPA attacks.

To enhance the protection against ML attacks, the hybrid APUF circuit is designed in this article. This mechanism incorporates the advantages of DAPUF and FF-APUF to enhance the randomness between the output response and the input challenge. As shown in Table 3, considering the stability of the APUF circuit, the effectiveness of protection against ML attacks, and the hardware resource overhead, we determine the feed-forward loop, as shown in Figure 6b. This circuit has four signal paths. Each path passes through 32 switch units and has four feed-forward loops. As shown in Table 5, under the premise of guaranteeing stability, randomness, and uniqueness, the proposed hybrid APUF can reduce the response prediction accuracy to less than 60% compared to existing APUFs, indicating a strong resistance to ML attacks.

To solve the problem of the existing methods being difficult to detect whether the jump directions of branches have been tampered with, the proposed CFI protection mechanism integrates the jump conditions of branches into the label calculation process. To reduce the storage overhead, the reference information size of this mechanism is basically the same as our previously proposed methods [10,13]. The Address_BB_ are all the lower 16 bits of BB addresses, and the integrity labels are all 16 bits. As shown in Table 6, this mechanism can achieve comprehensive protection of instruction codes, jump addresses, and jump directions. As shown in Table 8 and Table 9, compared to the previously proposed CFI protection methods [10,13], the method proposed in this article not only provides more comprehensive protection but also has lower performance overhead.

### 6.2. Limitations and Future Directions

It is undeniable that the proposed method in this article also has limitations. Attackers can not only obtain branch execution status through BPA attacks but also achieve the same attack goal through specter vulnerabilities in the processor cache [16]. This method proposed in this article focuses on branch prediction protection, which makes it difficult to defend against other types of Spectre attacks. The hybrid APUF circuit designed in this paper is mainly oriented to the lightweight encryption and decryption task of branch information, and the structure is relatively simple compared to the current general-purpose type of PUF circuits [38], and it is difficult to defend against more sophisticated ML attacks. The method proposed in this article can effectively monitor the CFI of direct jumps and conditional branches, but it is difficult to accurately detect attacks on indirect jumps. This is because there are only two jump directions of a branch, but an indirect jump has more unpredictable jump directions.

Therefore, our future directions will focus on the limitations of the proposed method. We will delve into the attacks and protection methods based on other Spectre vulnerabilities, not only the attacks and protection against branch prediction. We will continue to investigate confidentiality protection methods for the control flow of critical information, especially the PUF circuits with higher security. We will investigate the security methods for indirect jumps to achieve more comprehensive and fine-grained CFI protection.

## 7. Conclusions

This article proposes a hardware security protection method for conditional branches of embedded systems. This method calculates the number of BTB_Updates every 80 clock cycles during program execution. If the value exceeds the lock threshold, the lock field of each BTB entry is set to 1. At this point, any process will be unable to modify the branch information within BTB, making it difficult for attackers to obtain the execution status of program branches through attacks such as BPA. Moreover, to protect the confidentiality of branch information within BTB and BHT, this article designs a dynamic isolation mechanism based on hybrid PUF, which performs lightweight encryption and decryption on the indexes, branch addresses, and jump addresses of branches to prevent attackers from stealing these critical data. In order to resist machine learning attacks against APUF (such as CMA-ES, LR, and ANN), this article integrates the advantages of multiple APUF circuits, ensuring the stability, randomness, and uniqueness of the circuit while significantly reducing the accuracy of attackers in predicting the output responses. Once the above protection mechanisms are breached or bypassed, attackers can tamper with the instruction codes, jump addresses, or jump directions. The hardware SMU designed in this article will detect such attacks in a timely manner by comparing dynamic and static labels, achieving real-time monitoring of the control flow integrity. After FPGA implementation, this article evaluates the security, resource overhead, and performance overhead of the proposed method. The experimental results show that the hardware locking mechanism proposed in this paper can effectively resist BPA attacks, the designed hybrid APUF circuit has high resistance to ML attacks, and the proposed CFI monitoring mechanism can timely detect tampering attacks on instruction codes, jump addresses, and jump directions. Moreover, the resource overhead of the proposed method is about 5.4% of the SoC based on E906, and the performance overhead is less than 5.5%, making it suitable for lightweight embedded processors.

## Figures and Tables

**Figure 1 micromachines-15-00760-f001:**
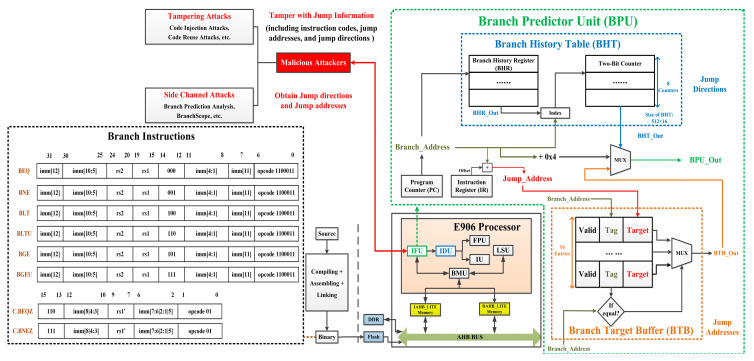
The security threats considered in this article.

**Figure 2 micromachines-15-00760-f002:**
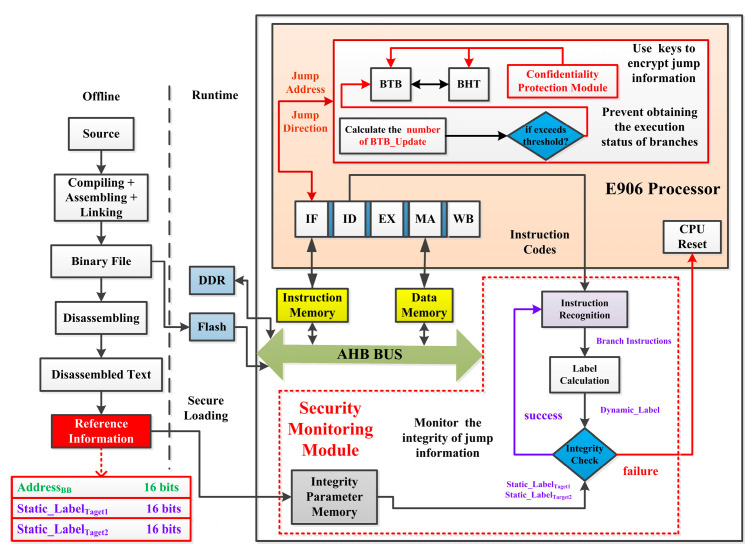
The overall structure of the proposed method.

**Figure 3 micromachines-15-00760-f003:**
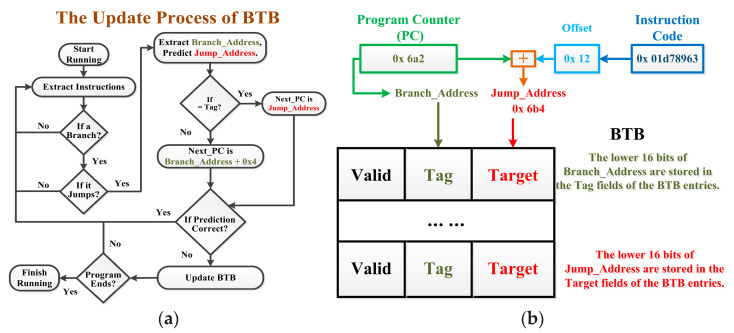
An example of the BTB update process. (**a**) The update process of BTB. (**b**) The example of writing new data for BTB.

**Figure 4 micromachines-15-00760-f004:**
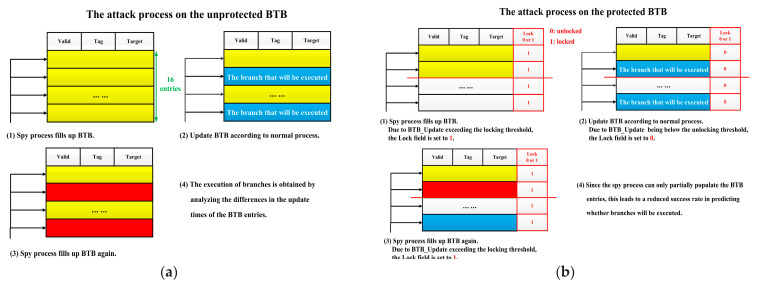
The principle of hardware locking mechanism against BPA Attacks. (**a**) The attack process on the unprotected BTB. (**b**) The attack process on the protected BTB.

**Figure 5 micromachines-15-00760-f005:**
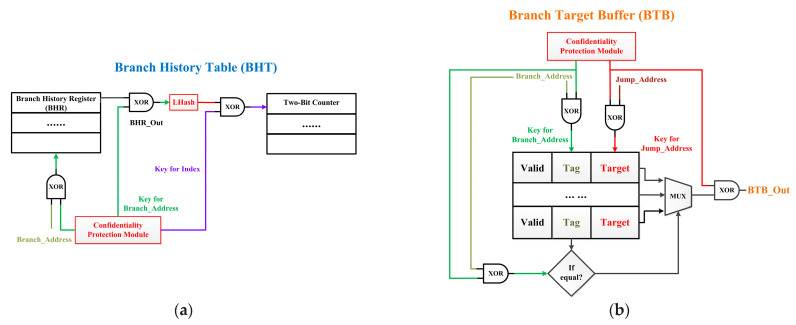
The principle of dynamic isolation mechanism proposed in this article. (**a**) The confidentiality protection for BHT. (**b**) The confidentiality protection for BTB.

**Figure 6 micromachines-15-00760-f006:**
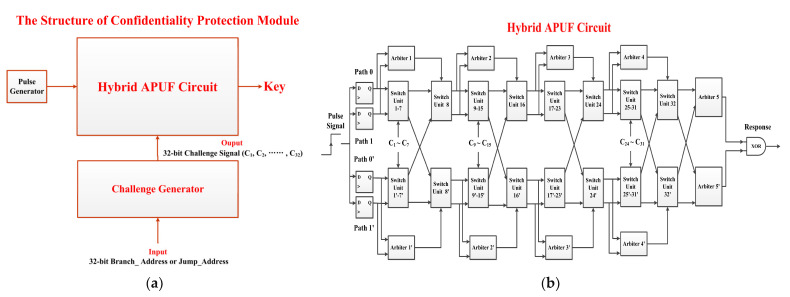
The main components of the confidentiality protection module. (**a**) The structure of confidentiality protection module. (**b**) The circuit of hybrid APUF.

**Figure 7 micromachines-15-00760-f007:**
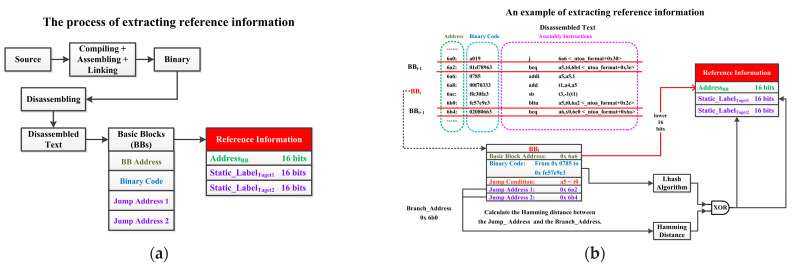
The extraction process of the reference information. (**a**) The process of extracting reference information. (**b**) The example of extracting reference information.

**Figure 8 micromachines-15-00760-f008:**
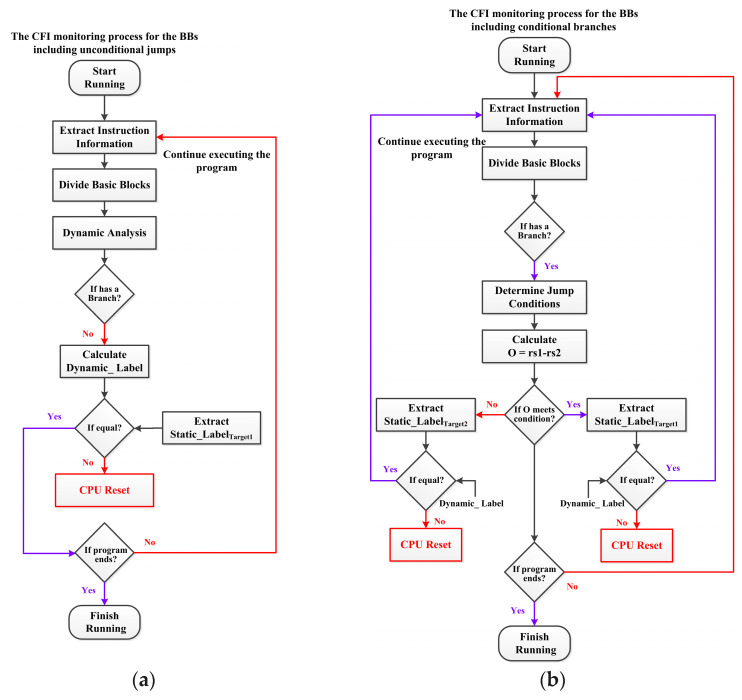
The CFI monitoring process. (**a**) The process for the BBs, including unconditional jumps. (**b**) The process for the BBs, including conditional branches.

**Table 1 micromachines-15-00760-t001:** Characteristics of the selected benchmarks.

Benchmarks	Proportion of Branches in All Instructions	Occupancy Rate of BTB	Max Number of BTB_Update
hello_world	12.54%	37.5%	6
coremark	15.41%	100%	8
basicmath	12.52%	37.5%	2
SHA1	13.07%	37.5%	3
FFT	13.60%	62.5%	5
bitcount	13.15%	50%	3
CRC16	13.23%	37.5%	3
patricia	13.04%	87.5%	7
quicksort	13.22%	62.5%	5
blowfish	13.21%	25%	3

**Table 2 micromachines-15-00760-t002:** The impact of Threshold_Lock_ and Threshold_unLock_ on the protection effectiveness and performance overhead.

Threshold_Lock_ and Threshold_unLock_	Attack Success Rate	Performance Overhead
Threshold_Lock_ is 9	55.4%	4.50%
Threshold_Lock_ is 10	56.1%	4.42%
Threshold_Lock_ is 11	56.9%	4.31%
Threshold_Lock_ is 12	57.1%	4.23%
Threshold_Lock_ is 13	62.3%	4.19%
Threshold_Lock_ is 14	69.8%	4.15%
Threshold_Lock_ is 15	74.6%	4.08%
Threshold_Lock_ is 16	83.6%	3.95%
Threshold_unLock_ is 11	64.4%	3.10%
Threshold_unLock_ is 10	63.1%	3.16%
Threshold_unLock_ is 9	61.9%	3.28%
Threshold_unLock_ is 8	61.0%	3.39%
Threshold_unLock_ is 7	60.3%	3.48%
Threshold_unLock_ is 6	59.6%	3.56%
Threshold_unLock_ is 5	59.2%	3.67%
Threshold_unLock_ is 4	58.6%	3.81%
Threshold_unLock_ is 3	58.1%	3.93%
Threshold_unLock_ is 2	57.6%	4.06%

**Table 3 micromachines-15-00760-t003:** The variation in PUF circuit indicators with the number of feed-forward loops.

Num_FF_	Stability	Accuracy ^1^	Resource ^2^
2	95.1%	58.8%	209
4	98.8%	53.3%	217
8	92.5%	60.1%	233
16	82.3%	64.9%	265

^1^ Based on the CMA-ES [38], which is one of ML algorithms, we predicted whether the output response of the proposed hybrid PUF circuit is correct. The number of CRPs used for training is 100 K. ^2^ Based on the Xilinx Kintex 7 FPGA (Alinx, Shanghai, China), we implemented the proposed hybrid PUF circuit. The basic hardware resources are look-up tables (LUTs).

**Table 4 micromachines-15-00760-t004:** Security testing for defending against Branch Prediction Analysis attacks.

Benchmarks	Attack Success Rate	
	Without Protection	With Hardware Locking BTB
hello_world	98.1%	58.3%
coremark	97.4%	59.6%
basicmath	97.9%	56.4%
SHA1	97.6%	57.5%
FFT	98.3%	58.1%
bitcount	97.3%	59.4%
CRC16	97.5%	58.7%
patricia	98.8%	57.9%
quicksort	98.6%	58.3%
blowfish	98.5%	55.9%

**Table 5 micromachines-15-00760-t005:** The security comparison of the multiple APUF circuits.

APUF Types	Stability	Randomness	Uniqueness	Accuracy		
				CMA-ES	LR	ANN
Traditional APUF	98.85%	51.31%	49.57%	98.28%	99.31%	99.27%
2XOR-APUF	98.83%	48.89%	49.65%	95.18%	97.32%	96.98%
DAPUF	98.81%	49.63%	50.11%	64.84%	65.65%	75.16%
FF-APUF	97.84	50.58%	48.91%	86.13%	87.65%	88.01%
Hybrid APUF	98.79%	49.85%	50.06%	53.32%	56.78%	57.93%

**Table 6 micromachines-15-00760-t006:** The detection rate of control flow tampering attacks.

Benchmarks	Attacks on Conditional Branches			Attacks on Unconditional Jumps	
	Instruction Codes	Jump Addresses	Jump Directions	Instruction Codes	Jump Addresses
hello_world	100%	100%	100%	100%	93.4%
coremark	100%	100%	100%	100%	92.2%
basicmath	100%	100%	100%	100%	95.7%
SHA1	100%	100%	100%	100%	96.3%
FFT	100%	100%	100%	100%	96.4%
bitcount	100%	100%	100%	100%	94.3%
CRC16	100%	100%	100%	100%	95.4%
patricia	100%	100%	100%	100%	96.8%
quicksort	100%	100%	100%	100%	96.1%
blowfish	100%	100%	100%	100%	93.6%

**Table 7 micromachines-15-00760-t007:** Comparison of the hardware resource overhead for confidentiality protection circuits.

Circuits	LUTs	Flip Flops	Slices
Traditional APUF	409	156	133
2XOR-APUF	672	165	223
DAPUF	687	165	228
FF-APUF	429	150	142
Hybrid APUF	712	168	231

**Table 8 micromachines-15-00760-t008:** The change in CPI caused by the proposed method.

Benchmarks	With Proposed Method	Without Proposed Method	Performance Overhead
hello_world	1.28	1.24	4.03%
coremark	3.38	3.21	5.30%
basicmath	2.37	2.28	3.95%
SHA1	2.07	1.98	4.55%
FFT	2.51	2.39	5.02%
bitcount	1.42	1.36	4.41%
CRC16	1.65	1.58	4.43%
patricia	1.55	1.47	5.44%
quicksort	1.83	1.76	3.98%
blowfish	3.51	3.35	4.78%

**Table 9 micromachines-15-00760-t009:** The comparison of security, resource overhead, and performance overhead.

Methods	Prevent Obtaining Execution Status of Branches	Encrypt Branch Information	Monitoring the CFI	Resource Overhead	Performance Overhead
Conditional Speculation [16]	Yes	No	No	About 1.52 KB	The average is 2.8%, but in extreme cases, it may approach 30%.
Secure Branch Predictor [17]	Yes	Yes	No	BTB_Area increased by 0.24%, PHT_Area increased by 0.11%.	The average is a few percent, but sometimes, it may be more than 20%.
Countermeasure against BPA [20]	Yes	No	No	Area increased by 8 KB	0.12%
HCIC [19]	No	Yes	Yes	Binary_Size increased by 0.78%.	0.95%
M-Cache based Security Monitor [10]	No	No	Yes	Area increased by 20.99%.	9.33%
Hardware Monitoring Module [13]	No	No	Yes	Consumed 486 Slices and 1374 LUTs.	Less than 9.52%
The Proposed Method	Yes	Yes	Yes	Consumed 613 Slices and 1851 LUTs.	Less than 5.5%

## Data Availability

Data are contained within this article.

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
