# Peer review of "A Hardware Security Protection Method for Conditional Branches of Embedded Systems"

_micromachines, 2024, doi:10.3390/mi15060760_

Round 1

Reviewer 1 Report

Comments and Suggestions for Authors

-The authors should refer the limitations of the proposed method

-The authors should add a discussion section which is a very important part of a research article or rename the conclusion as conclusion and discussion and enrich the section.

-The authors refer:The learning rate of this ANN is 0.01 and the optimizer is momentum algorithm. Why these settings were chosen? Why the authors have not tuned the network by testing different structures?

Comments on the Quality of English Language

-

Author Response

Dear Reviewer,

              Thank you for your review.

We have revised the manuscript based on your review comments.

(1) In response to your first comment, we add the Section 6.2, and discuss the limitations of the proposed method in this section. We have marked this section in red in the revised version of the manuscript.

(2) In response to your second comment, we add a discussion section. This section compares the proposed method with existing methods, analyzes the limitations of the proposed method, and proposes future research directions.

(3) As for your third comment, our research on machine learning attacks is just getting started, so at the moment we mainly refer to published literature and try to reproduce their attack methods. The ANN used in this paper is mainly based on Ref. [37] (Xu, C.; Zhang, L.; Law, M. -K.; Zhao, X.; Mak, P. -I.; Martins, R. P. Modeling Attack Resistant Strong PUF Exploiting Stagewise Obfuscated Interconnections With Improved Reliability. IEEE Internet of Things J. 2023, 10, 16300 - 16315.).The parameters of this ANN are also all in line with this literature.

Reviewer 2 Report

Comments and Suggestions for Authors

 This study proposes a hardware security protection method for conditional branches of embedded systems. This method calculates the number of updates to the branch target buffer (BTB). When it exceeds the threshold, BTB is locked to prevent attackers from analyzing the execution status of branches based on the time difference. Moreover, the hybrid physical unclonable function (PUF) circuit is designed to provide confidentiality protection for the jump directions, jump addresses, and their indexes, preventing attackers from stealing these critical data.

This is a topic of interest and for authors of this paper, I have the following comments.

1. The expression description of the paper needs improvement.  There are certain spelling mistakes and formatting inconsistency, please check the full manuscript carefully and try to improve the clarity of text. The introduction section should be improved by clarifying the similarities and differences between the related work and the proposed method. It is recommended to present research gap (in the literature) and highlight the contributions in the form of bullets.

2. The used security protection techniques should be justified and specify the implementation details/parameters of the used methods.

3. The text is not readable in Figures 3 to 8. Also too much information is displayed on these Figures without any description. It is recommended that authors simplify these Figures and provide a clear description of findings/illustrations displayed on these Figures.

4. Properly support conclusions in the discussion section by providing findings and figures/values. What's the influence of the parameters in the proposed method? I suggest author to give the discussion of the main parameters.

5. Also present limitations and future directions.

Comments on the Quality of English Language

Minor editing of English language required.

Author Response

Dear Reviewer,

       Thank you for your review.

We have revised the manuscript based on your review comments.

(1) In response to your first comment, we have corrected the spelling errors and introduction section of this article. We emphasize the similarities and differences between the proposed method and the existing methods, illustrated the problems of the existing methods, and highlighted the contributions of this article in the form of bullets.

(2) In response to your second comment, we specify the implementation details and parameters of the proposed method. We mark this portion of the text in red and add annotations to the revised version of the manuscript.

(3) In response to your third comment, we have split and simplified the images from Figure 3 through Figure 8 and added clear descriptions of these images.

(4) In response to your fourth comment, in the discussion section, we add the impact of parameters on the security, performance overhead, and resource overhead of the proposed method. Furthermore, in Section 4, we analyze the impact of the main parameters on the proposed methodology. For example, Table 2 in Section 4.1.2 shows the impact of ThresholdLock and ThresholdunLock on the protection effectiveness and performance overhead.

(5) In response to your fifth comment, we add the Section 6.2, and discuss the limitations of the proposed method and the future directions in this section.

Reviewer 3 Report

Comments and Suggestions for Authors

This work is interesting.

3 methods are suggested to protect branch prediction units.

To find attacks that attempting to fill all BTB entries, the threshold of BTB_Update every 80 clock cycles is designed.

to resist ML attacks, a hybrid APUF circuit deepening the circuit nonlinearity is suggested.

to check and protect whether the jump directions are correct, dynamic and static labels are compared.

These methods are implemented on FPGA, so small resource overhead are achieved.

I have some concerns:

Minor issue:

More details may be given in the abstract to show the key points of these 3 methods clearly.

A photo of the experimental system may be given.

Author Response

Dear Reviewer,

       Thank you for your review.

We have revised the manuscript based on your review comments.

(1) In response to your first comment, we revise the abstract to show more details of the proposed method.

(2) As for your second comment, the photographs of the main experimental system of this paper are shown below. This system includes an FPGA development board, a Lenovo server, a display, a keyboard and mouse. We do not add this photo to the revised version of the manuscript, since the photo simply shows the components of the experimental system.

Round 2

Reviewer 2 Report

Comments and Suggestions for Authors

Authors have incorporated the recommended changes in the revised version of paper.

Comments on the Quality of English Language

 Minor editing of English language required